# Analysis of the Roles of the *ISLR2* Gene in Regulating the Toxicity of Zearalenone Exposure in Porcine Intestinal Epithelial Cells

**DOI:** 10.3390/toxins14090639

**Published:** 2022-09-16

**Authors:** Zhenbin Bi, Xuezhu Gu, Yeyi Xiao, Yajing Zhou, Wenbin Bao, Shenglong Wu, Haifei Wang

**Affiliations:** 1Key Laboratory for Animal Genetics, Breeding, Reproduction and Molecular Design, College of Animal Science and Technology, Yangzhou University, Yangzhou 225009, China; 2Lvliang Central Animal Husbandry and Veterinary Station, Huaian 211600, China; 3Joint International Research Laboratory of Agriculture & Agri-Product Safety, Yangzhou University, Yangzhou 225009, China

**Keywords:** pig, zearalenone, *ISLR2* gene, cytotoxicity, molecular target

## Abstract

Zearalenone (ZEN) is one of the mycotoxins that pose high risks for human and animal health, as well as food safety. However, the regulators involved in ZEN cellular toxicity remain largely unknown. Herein, we showed that cell viability of porcine intestinal epithelial cells (IPEC-J2) tended to decrease with increasing doses of ZEN by the cell counting kit-8 assay. Expression of the *ISLR2* (immunoglobulin superfamily containing leucine-rich repeat 2) gene in IPEC-J2 cells was significantly downregulated upon ZEN exposure. Furthermore, we found the dose–effect of ZEN on *ISLR2* expression. We then overexpressed the *ISLR2* gene and observed that overexpression of *ISLR2* obviously reduced the effects of ZEN on cell viability, apoptosis rate and oxidative stress level. In addition, *ISLR2* overexpression significantly decreased the expression of *TNF-α* and *IFN-α* induced by ZEN. Our findings revealed the effects of ZEN on the *ISLR2* gene expression and indicated the *ISLR2* gene as a novel regulator of ZEN-induced cytotoxicity, which provides potential molecular targets against ZEN toxicity.

## 1. Introduction

Mycotoxins are highly toxic secondary products produced by fungi species such as Aspergillus, Penicillium, and Fusarium. The toxic syndrome caused by the ingestion of mycotoxins in humans and animals is called mycotoxicosis [1,2]. Mycotoxicity causes the development of cancer, mutations and malformations, as well as toxic reactions in cells, nerves and kidneys and other guts [1]. Several mycotoxins are particularly important for public health and agronomic development, including aflatoxin, ochratoxin A, fumonisin, deoxynivalenol and zearalenone (ZEN) [2]. ZEN is a non-steroidal estrogenic mycotoxin [3]. From a structural point of view, ZEN has a lactone ring, a feature that allows it to be heat resistant up to 150 °C [4], and thus it shows high stability during both post-processing and storage. This characteristic of ZEN stability at increased temperature and pressure makes it detectable in fresh plants and processed products [5,6]. The non-degradability of mycotoxins at high temperature and pressure results in great potential harm of mycotoxin contamination for food security.

Among the farm animals, pigs are the most significantly affected species and are much more sensitive to zearalenone than rodents and other farm animals. The intestine is the main organ that absorbs ZEN. Ingestion of zearalenone-contaminated feed by livestock can cause damage to their intestinal function and integrity, which causes the occurrence of diseases. ZEN induces apoptosis and oxidative stress in porcine intestinal epithelial cells, and these cellular damages induced by ZEN are closely linked to several key signaling pathways; it has been shown that glutamine and DL-selenomethionine play a mitigating role in ZEN-induced apoptosis and oxidative stress through PI3K/Akt and Nrf2/Keap1 signaling pathways, respectively [7,8]. According to previous studies, ZEN mainly causes structural damage in the porcine jejunum, inducing inflammation and apoptosis of jejunal epithelial cells [9,10,11]. However, the underlying mechanism of ZEN-induced jejunal epithelial injury is largely unknown.

*ISLR2* (immunoglobulin superfamily containing leucine-rich repeat 2) is a member of the leucine-rich repeat (LRR) and immunoglobulin (LIG) family of membrane proteins, which is preferentially expressed in the central and peripheral nervous systems [12]. It is surprising that few molecules contain both sequence elements. In contrast, the nature of the LRR motif in the *ISLR2* gene is important for generating various interactions with exogenous factors in the immune system and with a large number of different cell types in the developing nervous system [13]. Both intracellular and extracellular LRR proteins, are well characterized in the natural immune system, from plants to mammals [14]. Interactions of *ISLR2* with ret proto-oncogene was involved in regulating the development of the gastrointestinal tract, and alterations of *ISLR2* expression levels were found to be associated with pseudoexfoliation syndrome [12,15]. We previously found that the expression of *ISLR2* was significantly changed in porcine intestinal epithelial cells upon ZEN exposure [16]. However, the regulatory mechanisms and functions of *ISLR2* in response to ZEN exposure remain unknown.

In this study, we explored the effects of ZEN on the expression of the *ISLR2* gene in porcine intestinal epithelial cells (IPEC-J2). In addition, we investigated the roles of the *ISLR2* gene in regulating the cytotoxic effects induced by ZEN. Our findings will provide insight into the roles of the *ISLR2* gene in cellular responses to ZEN exposure and enhance our understanding of the molecular mechanisms involved in the toxicological processes of ZEN.

## 2. Results

### 2.1. Dose-Effect of ZEN on Cell Viability and ISLR2 Expression

We first determined the optimal doses of ZEN on IPEC-J2 cells using cell viability analysis and found that the cell viability was significantly decreased (*p* < 0.01) and around 50% at a dose of 10 μg/mL compared with the control (Figure 1A). Meanwhile, we detected the expression of *ISLR2* by qPCR at different ZEN doses and found that the expression of *ISLR2* was significantly decreased (*p* < 0.01) with a higher dose of ZEN in comparison with the control (Figure 1B).

### 2.2. The Role of ISLR2 in ZEN Induced Cytotoxicity and Oxidative Stress

To investigate the role of *ISLR2*, we constructed the plasmids overexpressing *ISLR2*. The recombinant plasmid containing the *ISLR2* coding sequence was confirmed by agarose gel analysis (Appendix A) and DNA sequencing, which shows consistent sequences with the known sequences of the *ISLR2* coding region (Appendix A). Expression analysis indicated that *ISLR2* expression in the overexpression group was more than 900-fold higher than that in the control group (*p* < 0.01) (Appendix A). To investigate the functions of *ISLR2* in response to ZEN-induced toxicity, we analyzed cell viability and ROS level in cells overexpressing *ISLR2*. The results showed that the cell viability of ZEN-treated cells was significantly decreased compared with the control group, while the cell viability in cells overexpressing *ISLR2* was significantly higher than that of the ZEN exposure group (Figure 2A). In addition, after ZEN treatment, ROS levels in cells overexpressing *ISLR2* were significantly lower than that of control cells (*p* < 0.01) (Figure 2B), indicating the involvement of *ISLR2* in mediating ZEN-induced ROS production.

### 2.3. Role of ISLR2 in ZEN-Induced Cell Apoptosis and Inflammation

To further investigate the roles of *ISLR2* in ZEN-induced toxicity, we detected the apoptotic level of cells exposed to ZEN by Annexin V-APC/PI. The results showed that the total apoptotic cells increased after ZEN exposure. The apoptosis rate in cells overexpressing *ISLR2* was significantly lower than that of control cells (*p* < 0.01) (Figure 3A,B).

Expression analysis of apoptosis-related proteins showed that the expression of *Caspase3*, *Caspase9,* and *BAX* in cells overexpressing *ISLR2* was significantly lower than that of the control cells (Figure 3C) (*p* < 0.01). Western blot assay further confirmed the expression of these proteins (Figure 3D). Furthermore, we detected the expression of proinflammatory cytokines (*TNF-α*, *IL-6*) and interferon *IFN-α*. The results showed that *ISLR2* overexpression significantly decreased the expression of *TNF-α* and *IFN-α* induced by ZEN (*p* < 0.01), while the *IL-6* expression level did not change significantly (Figure 3E).

## 3. Discussion

ZEN exerts different mechanisms of toxicity in different cell types at different doses, which leads to estrogen-like effects at low doses and cell death at high doses [17]. Previous studies showed that ZEN induces *HSP70* expression in a time and dose-dependent manner in HEPG2 cells [18]. Under different doses of DON treatment, DON inhibited expression of the *TEM8* gene that plays important roles in cell migration, and the inhibitory effects of DON on cell migration could be reduced by overexpression of *TEM8* [19]. In this study, we revealed the dose–effect of ZEN on *ISLR2* expression in IPEC-J2 cells. These findings indicated that the expression of some genes is responsive to mycotoxin doses, and these genes may be important potential regulators involved in mycotoxin toxicological processes.

ZEN exposure can result in an imbalance of oxidative and antioxidant effects, allowing the massive production of free radicals. The massive accumulation of free radicals leads to the destruction of DNA, proteins, and lipids [20]. The accumulation of excessive ROS is one of the causes of apoptosis. Previous studies have reported that *SelS* overexpression mitigates OTA-induced cytotoxicity and apoptosis [21]. In addition, overexpression of *HO-1* reduces the DON-induced ROS and DNA damage by maintaining DNA repair, antioxidant activity, and autophagy [22]. In this study, we observed that *ISLR2* overexpression significantly increased the cell viability and reduced the ROS levels of cells exposed to ZEN, indicating the potential of *ISLR2* as a molecular target for controlling ZEN-induced toxicity.

After *ISLR2* gene overexpression, the toxicity of ZEN in IPEC-J2 cells could be inhibited. It has been reported that *ISLR2* is composed of two parts: leucine-rich repeat (LRR) and immunoglobulin (LIG) families, while LRR in them plays important roles in the innate immune system [13], indicating that the function of *ISLR2* may be related to immunity. According to relevant reports, the loss of *ISLR* in stromal cells significantly impaired the regeneration of the intestine and inhibited the development of colon tumors [23]. In non-small cell lung cancer (*NSCLC*), silencing *ISLR* increased the apoptotic rate of cells [24]. In chicken myoblasts, silencing *ISLR* similarly caused significantly higher expression of apoptosis-related proteins such as *Caspase3*, *Caspase8*, and *Caspase9* [25]. These results suggested that inhibition of *ISLR* expression may be detrimental to cell proliferation, and we therefore speculate that *ISLR2* may play a similar role as a paralogous homolog of *ISLR*. Combined with the analysis of our results, we suggest that high expression of *ISLR2* contributes to increased cellular resistance to ZEN through oxidative stress and apoptosis. Herein, our data showed that high expression of *ISLR2* decreases the expression of these inflammatory factors, suggesting that *ISLR2* may play a role in the inflammatory response. Taken together, our results suggest that *ISLR2* alleviated the damage caused by ZEN by reducing apoptosis and alleviating inflammatory responses.

## 4. Conclusions

In conclusion, we found the dose–effect of ZEN on the expression of the *ISLR2* gene and explored the functions of *ISLR2* expression in mitigating the cytotoxicity induced by ZEN exposure. Findings of this study shed light on the roles of *ISLR2* in IPEC-J2 cells upon ZEN exposure and provided potential molecular targets for protection against ZEN cytotoxicity.

## 5. Materials and Methods

### 5.1. Construction of ISLR2 Overexpression Vector

The CDS region of *ISLR2* was amplified by PCR with the addition of restriction of endonucleases digestion sites. A 50 μL reaction system consisted of 200 ng cDNA, 1.5 µL each of forward and reverse primers (10 µmol/L), 25 µL 2×PCR Buffer, 10 µL 2 mM dNTPs, 1.0 µL KOD FX (1.0 U/µL) mixture, and ddH2O (50 µL total) (Toyobo, Osaka, Japan). PCR reactions were as follows: 94 °C for 2 min, 40 cycles of 98 °C for 10 s, 68 °C for 2 min. PCR products were double digested and then ligated into the linear pcDNA3.1 plasmid using T4 DNA ligase at 16 °C overnight (Vazyme, Nanjing, China). PCR products were confirmed by agarose gel and Sanger sequencing. The plasmids expressing *ISLR2* were transfected into cells using jetPRIME (Polyplus, Illkirch, France) following the manufacturer’s protocols. The primer information is shown in Table 1.

### 5.2. Cell Culture

IPEC-J2 cells were plated into 6-well and 12-well plates and incubated in DMEM medium containing 1% penicillin streptomycin (1 mg/mL) and 10% FBS at 37 °C and 5% CO_2_ in an incubator.

### 5.3. Cell Viability Assay

Cells were plated into 96-well plates at a density of 2000 cells per well. After overnight incubation, cells were treated with different concentrations (0, 1, 5, 10, 20, 40 μg/mL) of ZEN and incubated for 48 h. Cell viability was measured using CCK8 reagents following the manufacturer’s protocols. The absorbance at 450 nm was quantified on a Tecan Infinite 200 microplate reader (Tecan, Männedorf, Switzerland).

### 5.4. Cell Apoptosis Detection

Cells were plated on six-well plates and exposed to 10 μg/mL of ZEN for 48 h. Cell samples were washed with PBS and sequentially stained with Annexin V-APC and (PI) using the Annexin V-APC/PI Apoptosis Kit (Elabscience, Wuhan, China). The stained samples were analyzed by flow cytometry and followed by CytExpert 2.3 (Beckman Coulter, Brea, CA, USA) to count cell apoptosis rate.

### 5.5. Determination of Oxidative Stress Index

Cells were plated on 6-well plates and exposed to 10 μg/mL of ZEN for 48 h. Cell samples were washed twice with pre-chilled PBS and then incubated with 10 µM DCHF-DA for 30 min at 37 °C. ROS levels in cells were measured with Reactive Oxygen Species Analysis Kit (Solarbio, Beijing, China) and analyzed by flow cytometry (Beckman Coulter, Brea, CA, USA).

### 5.6. qRT-PCR

Total RNA was isolated using the Trizol method. Reverse transcription was performed with the HiScript II Q Select RT SuperMix for qPCR kit (Vazyme, Nanjing, China). The cDNA was used as the template. The reactions were performed on a real-time PCR machine (ABI Step One Plus, Pleasanton, CA, USA). The GAPDH gene was used as the housekeeping gene. Gene relative expression was calculated by the 2^−ΔΔCT^ method [26]. The primer sequences of each gene are shown in Table 1.

### 5.7. Western Blot Analysis

Cells were washed twice with pre-cooled PBS, mixed with RIPA lysate (Applygen, Beijing, China) and protease inhibitor (CWBIO, Beijing, China). Total proteins were isolated by utilizing a protein extraction reagent, denatured by boiling for 10 min, separated with SDS-PAGE, and transferred to PVDF membranes. Blocking of the membrane was performed with 5% non-fat dry milk and 0.2%Tween for 2 h at room temperature. Proteins were incubated with antibodies, including anti-*Cleaved-Caspase3*, anti-*Cleaved-Caspase9*, anti-*BAX* and anti-*HSP90,* overnight at 4 °C. Incubation with secondary antibody for 2 h was then conducted at room temperature. Finally, the bands were detected by ECL.

## Figures and Tables

**Figure 1 toxins-14-00639-f001:**
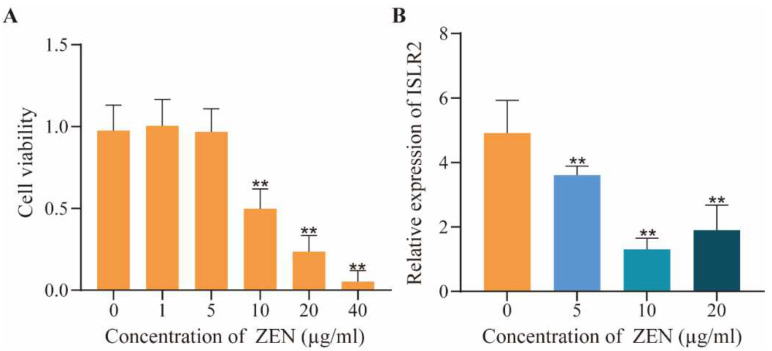
Effects of ZEN on cell viability and *ISLR2* expression. (**A**) Detection of cell viability of cells exposed to different doses (0, 1, 5, 10, 20, 40 μg/mL) of ZEN. (**B**) *ISLR2* expression changes in cells exposed to different doses (0, 5, 10, 20 μg/mL) of ZEN for 48 h. Different color of bars represents different concentration of ZEN. Data are presented as mean ± SD. Significance compared with control, ** *p* < 0.01.

**Figure 2 toxins-14-00639-f002:**
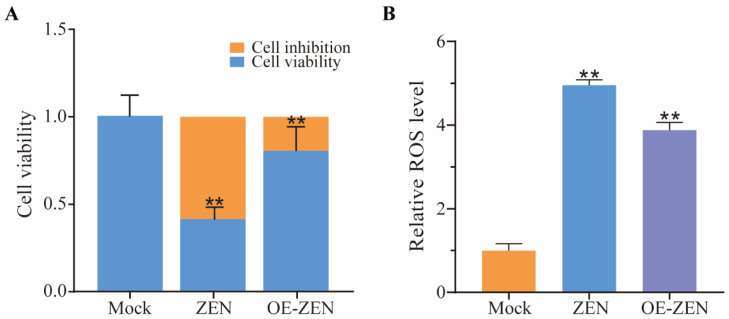
Effects of *ISLR2* overexpression on ZEN-induced cytotoxicity and oxidative stress. (**A**) Detection of cell viability in the Mock, ZEN, and OE-ZEN groups. (**B**) Relative ROS levels in the Mock, ZEN, and OE-ZEN groups. Mock, cells without ZEN treatment; ZEN, cells exposed to 10 μg/mL ZEN; OE-ZEN, cells overexpressing *ISLR2* and exposed to 10 μg/mL ZEN. Data are presented as mean ± SD. Significance compared with control, ** *p* < 0.01.

**Figure 3 toxins-14-00639-f003:**
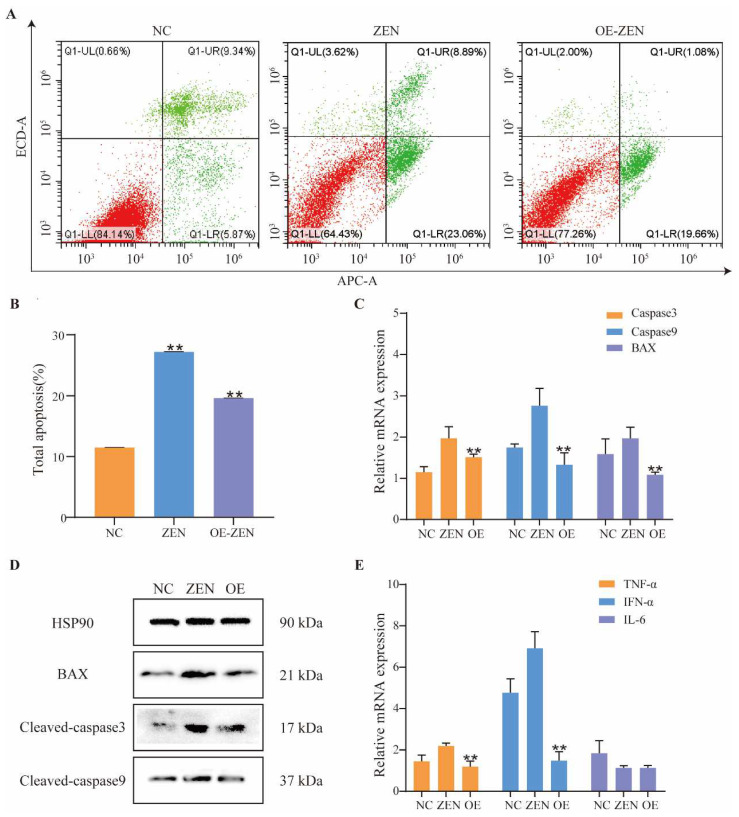
*ISLR2* overexpression alleviates ZEN-induced cell apoptosis and inflammation. (**A**) Cell apoptosis rate determined by annexin V-APC/PI staining using flow cytometry. Red and green color indicate the percentage of normal cells and apoptotic cells in the total cells, respectively. (**B**) Quantification of cell apoptosis rate in NC, ZEN, and OE-ZEN groups. (**C**) Relative expression of *Caspase3*, *Caspase9* and *BAX* in the NC, ZEN, and OE groups. (**D**) Protein expression of *Cleaved-Caspase3*, *Cleaved-Caspase9,* and *BAX* in the NC, ZEN and OE groups. (**E**) Relative expression of *TNF-α*, *IFN-α*, and *IL-6* in the NC, ZEN, and OE groups. NC, cells without ZEN treatment; ZEN, cells exposed to 10 μg/mL ZEN; OE-ZEN, cells overexpressing *ISLR2* and exposed to 10 μg/mL ZEN. Data are presented as mean ± SD. Significance compared with control, ** *p* < 0.01.

**Table 1 toxins-14-00639-t001:** Primer sequences used in qRT-PCR and PCR assays.

Gene	Sequence (5′–3′)	Product Length
*GAPDH*	F:GGTCGGAGTGAACGGATTT R:ATTTGATGTTGGCGGGAT	245 bp
*ISLR2*	F:GGTCCAAGGCGAGGTTGC R:CGAACTGATGCGCGTACTTG	242 bp
*Caspase3*	F:GGAATGGCATGTCGATCTGGT R:ACTGTCCGTCTCAATCCCAC	351 bp
*Caspase9*	F:TGGAACTCAAGCCAGAGGAG R:CTGCATTCAGGACGTAAGCC	195 bp
*BAX*	F:TGCCTCAGGATGCATCTACC R:AAGTAGAAAAGCGCGACCAC	199 bp
*TNF-a*	F:TTCCAGCTGGCCCCTTGAGC R:GAGGGCATTGGCATACCCAC	146 bp
*IL-6*	F:TTCACCTCTCCGGACAAAAC R:TCTGCCAGTACCTCCTTGCT	122 bp
OE-*ISLR2*	F:ttAAGCTT*gccacc*ATGGGCTCCAGCCCAGAC R:caGAATTCTCAGCCTGCTGTCTGCCTATAG	2431 bp
*ISLR2*-SNP	F:CTGGGTTTAGGTATGTTAGA R:ACACTGGCTCAGGACTC	419 bp

Note: For the primer of OE-*ISLR2*, the underlined letters indicate the restriction enzyme cutting sites, the italic is the Kozak sequence, and the lowercase is the protected base of the restriction enzyme cutting site.

## Data Availability

Not applicable.

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
