# Peer review of "Analysis of the Roles of the ISLR2 Gene in Regulating the Toxicity of Zearalenone Exposure in Porcine Intestinal Epithelial Cells"

_toxins, 2022, doi:10.3390/toxins14090639_

Round 1

Reviewer 1 Report

This article has described the results of a molecular biological study of the effects of ZEN on a small intestinal epithelial-derived cell line expressing ISLR2, based on previous studies showing significant changes in ISLR2 expression during ZEN exposure-induced transcriptome changes. The design of the experiments in this study is appropriate and the experiments are well performed; there are many unknowns regarding the biological role of ISLR2, and the results are novel and interesting. With that in mind, this reviewer requests that the authors consider again the following points.

Main points      

(1)  To determine whether a G/A mutation in the ISLR2 promoter region affects NF-kB binding or not, prediction based on binding sequence alone may not be sufficient. In my opinion, this claim of the authors requires practical evidence based on molecular biological experiments. For example, is the expression level of ISLR2 affected by the addition of NF-kB inhibitors? Although a somewhat classical approach, why not do a gel mobility shift assay with ISLR2 promoter sequence probes using IPEC-J2 cell lysates? As the cited reference 15 the authors stated as a previous study validates the transcriptome of a ZEN treated cell line, what about the expression profiles of NF-kB?

(2)  How does the G/A mutation in the ISLR2 promoter region affect sensitivity to ZEN exposures, and is it possible to compare cases of mycotoxicosis  in Meishan pigs and Large White pigs?    

Minor points     

(3)  Page 1, Line 36: When the intestine is exposed to mycotoxins, it activates antibacterial defense mechanisms, and reducing antibacterial effects are one of the mechanisms by which mycotoxins infect the intestine.   

 A citation is required.    

(4)  Section 3.1, Line 162, 164, 166, 168: Fig. 3    

It is assumed to be a typographical error that should be Fig. 1.   

(5)  Figure 1    

Fig. 1B does not seem to be necessary to present in the main text. I think Fig. 1C doesn't contain much point in presenting the results of the Sanger sequencer analysis, which is only a small part of the whole. Therefore, Fig. 1B and C could be presented as supplementary figures, not in the main text.

Author Response

This article has described the results of a molecular biological study of the effects of ZEN on a small intestinal epithelial-derived cell line expressing ISLR2, based on previous studies showing significant changes in ISLR2 expression during ZEN exposure-induced transcriptome changes. The design of the experiments in this study is appropriate and the experiments are well performed; there are many unknowns regarding the biological role of ISLR2, and the results are novel and interesting. With that in mind, this reviewer requests that the authors consider again the following points.

Main points      

(1)  To determine whether a G/A mutation in the ISLR2 promoter region affects NF-kB binding or not, prediction based on binding sequence alone may not be sufficient. In my opinion, this claim of the authors requires practical evidence based on molecular biological experiments. For example, is the expression level of ISLR2 affected by the addition of NF-kB inhibitors? Although a somewhat classical approach, why not do a gel mobility shift assay with ISLR2 promoter sequence probes using IPEC-J2 cell lysates? As the cited reference 15 the authors stated as a previous study validates the transcriptome of a ZEN treated cell line, what about the expression profiles of NF-kB?

Response: Thanks for your suggestions. We have quantified the expression of ISLR2 after inhibition of NF-kB by the addition of NF-kB inhibitor and found that the expression level of ISLR2 was significantly decreased with addition of MG132. The results have been provided in Figure 5D. Transcriptome analysis of the cited reference 15 revealed that the expression of NF-kB was downregulated post ZEN exposure.

 (2)  How does the G/A mutation in the ISLR2 promoter region affect sensitivity to ZEN exposures, and is it possible to compare cases of mycotoxicosis in Meishan pigs and Large White pigs?    

Response: In this study, we found significant decreases in the expression of ISLR2 upon ZEN exposure, and ISLR2 overexpression reduced the cytotoxicity of ZEN. The G/A mutation may affect sensitivity to ZEN exposure by regulating the expression of ISLR2. It is possible to compare cases of mycotoxicosis in the two pig breeds by supplying dietary with ZEN to the pigs with different genotypes at this locus and further testing the toxic effects on animals. This is an interesting question, and it will be our further endeavors to investigate the effects of genetic variants on sensitivity to ZEN in different genotypic pigs.

Minor points     

(3)  Page 1, Line 36: When the intestine is exposed to mycotoxins, it activates antibacterial defense mechanisms, and reducing antibacterial effects are one of the mechanisms by which mycotoxins infect the intestine.   

 A citation is required.    

Response: As no relevant reports were found and this statement was not closely linked with our study, the statement has been deleted in the revised manuscript.

(4)  Section 3.1, Line 162, 164, 166, 168: Fig. 3

It is assumed to be a typographical error that should be Fig. 1.   

Response: Thanks for your suggestions. It is a typographical error and we have corrected it.

(5)  Figure 1    

Fig. 1B does not seem to be necessary to present in the main text. I think Fig. 1C doesn't contain much point in presenting the results of the Sanger sequencer analysis, which is only a small part of the whole. Therefore, Fig. 1B and C could be presented as supplementary figures, not in the main text.

Response: Thanks for your suggestions. We have placed Fig. 1B and Fig. 1C as supplementary figures in our revised manuscript.

Reviewer 2 Report

Review for

Article 

Analysis of the Roles of the ISLR2 Gene in Regulating the Toxicity of Zearalenone Exposure in Porcine Intestinal Epithelial Cells

Zearalenone (ZEN) is one of the mycotoxins that poses high risks for human and animal

health as well as food safety.

Zearalenone toxicity is a huge research area

1,030 document results

TITLE-ABS-KEY ( zearalenone  AND  toxicity ) 

And studies about the regulators involved in ZEN cellular toxicity are welcome

In this paper, authors found the significant decrease(s?) of the ISLR2 gene in porcine epithelial

cells upon ZEN exposure. They further overexpressed the ISLR2 gene and observed that overexpression of ISLR2 obviously reduced the toxic effects of ZEN on cell activity, apoptosis rate and oxidative

stress level.

----------------------------------

According to previous studies, ZEN mainly causes structural damage in the porcine jejunum, inducing inflammation and apoptosis of jejunal epithelial cells [9-11]. However, the underlying mechanism of ZEN-induced jejunal epithelial injury is largely unknown.

What is the final objective?

Mitigation of ZEN biological effects?

ZEN cannot be avoided in feed and food, with good production and storage practices?

------------------------

OK for ethics

2.1. Ethics Statement

The animal study proposal was approved by the Institutional Animal Care and Use

Committee (IACUC) of the Yangzhou University Animal Experiments Ethics Committee

[permit number: SYXK(SU)IACUC2012-0029]. All experimental methods were conducted

in accordance with the related guidelines and regulations.

---------------------------------

Very impressive, large study

One hundred Meishan pigs from Meishan Pig National Breeding Conservation Center in Taicang City (Jiangsu Province, China) and 300 Large White pigs from Changzhou

Kang Le Farming Company (Jiangsu Province, China) were used in this study.

-------------------------------

ZEN exposure can result in an imbalance of oxidative and antioxidant effects, allowing the massive production of free radicals. The massive accumulation of free radicals

leads to the destruction of DNA, proteins, and lipids [20]. The accumulation of excessive

ROS is one of the causes of apoptosis.

Mechanism specific to ZEN or common to other mycotoxins?

-----------------------------

Previous studies have reported that SelS overexpression alleviated OTA-induced cytotoxicity and apoptosis[21]. In addition, overexpression of HO-1 reduces the ROS and DNA damage caused by DON exposure through maintaining DNA repair, antioxidant activity, as well as autophagy[22].

+ your study

That means there is a research axis in the world about mycotoxin management through metabolic engineering?

--------------------------------------

Author Response

Analysis of the Roles of the ISLR2 Gene in Regulating the Toxicity of Zearalenone Exposure in Porcine Intestinal Epithelial Cells

Zearalenone (ZEN) is one of the mycotoxins that poses high risks for human and animal health as well as food safety. Zearalenone toxicity is a huge research area

1,030 document results

TITLE-ABS-KEY (zearalenone AND toxicity) 

And studies about the regulators involved in ZEN cellular toxicity are welcome.

In this paper, authors found the significant decrease(s?) of the ISLR2 gene in porcine epithelial cells upon ZEN exposure. They further overexpressed the ISLR2 gene and observed that overexpression of ISLR2 obviously reduced the toxic effects of ZEN on cell activity, apoptosis rate and oxidative stress level.

According to previous studies, ZEN mainly causes structural damage in the porcine jejunum, inducing inflammation and apoptosis of jejunal epithelial cells [9-11]. However, the underlying mechanism of ZEN-induced jejunal epithelial injury is largely unknown.

What is the final objective?

Mitigation of ZEN biological effects?

ZEN cannot be avoided in feed and food, with good production and storage practices?

Response: The final objective of this study was to explore the functions of the ISLR2 gene in regulating ZEN toxicity. We have reorganized this sentence in our revised manuscript.

According to previous studies, approximately 75% – 100% of the samples also have one or more mycotoxins, and at low doses can still affect the health of animals. ZEN are produced mainly due to improper storage of feed. Meanwhile, from a structural point of view, ZEN has a lactone ring, a feature that allows it to be heat resistant up to 150°C, and thus it shows high stability during both post-processing and storage. We can also avoid the negative impact of ZEN by managing sanitation, controlling insects, properly irrigating and fertilizing, keeping the storage environment dry and legislating for good production and storage. ZEN can be effectively controlled with good production and storage practices, while it cannot be fully avoided in the practical production process according to the reports on mycotoxin contamination.

OK for ethics

2.1. Ethics Statement

The animal study proposal was approved by the Institutional Animal Care and Use Committee (IACUC) of the Yangzhou University Animal Experiments Ethics Committee [permit number: SYXK(SU)IACUC2012-0029]. All experimental methods were conducted in accordance with the related guidelines and regulations.

 Very impressive, large study

One hundred Meishan pigs from Meishan Pig National Breeding Conservation Center in Taicang City (Jiangsu Province, China) and 300 Large White pigs from Changzhou Kang Le Farming Company (Jiangsu Province, China) were used in this study.

ZEN exposure can result in an imbalance of oxidative and antioxidant effects, allowing the massive production of free radicals. The massive accumulation of free radicals leads to the destruction of DNA, proteins, and lipids [20]. The accumulation of excessive ROS is one of the causes of apoptosis.

Mechanism specific to ZEN or common to other mycotoxins?

Response: Thanks for your comments. This is a property shared by mycotoxins. Mycotoxin exposure results in accumulation of ROS, which further causes cell apoptosis.

Previous studies have reported that SelS overexpression alleviated OTA-induced cytotoxicity and apoptosis [21]. In addition, overexpression of HO-1 reduces the ROS and DNA damage caused by DON exposure through maintaining DNA repair, antioxidant activity, as well as autophagy [22].

+ your study

That means there is a research axis in the world about mycotoxin management through metabolic engineering?

Inspired by these reports, mycotoxin-induced cytotoxicity and apoptosis-related pathways are closely influenced by gene expression levels.

Response: Thanks for your comments. Identification of key regulators is one of the efficient methods to provide potential molecular targets against mycotoxin toxicity. Therefore, there is emerging studies focusing on roles of gene expression in regulating mycotoxin toxicity, which indicates the research axis in the world about mycotoxin management through metabolic engineering.

Reviewer 3 Report

In this study, the authors investigated the effect of Zearalenone (ZEN) on porcine intestinal cells looking at is the regulators involved in ZEN cellular toxicity in this model. Authors demonstrated a significant decreases in expression of the ISLR2 gene suggesting a role of this gene in ZEN toxicity. Role of ISLR2 was further demonstrated by the authors as when they overexpressed the gene, so protection against ZEN was observed. Finally, looking at expression regulation of ISLR2, the authors found a G/A mutation located in the promoter region decreasing its transcription. Authors predicted that the G/A mutation is affecting the binding of NF-κB. Overall, this study identified ISLR2 gene as a potential novel target of ZEN-induced toxicity and how it can be regulated by NFkB.

I found the study interesting. However, I have few issues with this manuscript.

Few comments here :

1- in the abstract, the authors used ZEN and ZEA term. Please homogenize.

2- in the abstract (but please check the full manuscript) please correct « decrease of » to « decrease in ». same for « increase of » to « increase in »

3- In the abstract : please indicate in the abstract or title the meaning of ISLR2 (when first used : title or first use in abstract)

4- in the abstract : please indicate that the porcine intestinal cells used are IPEC-J2.

5- introduction : Although present in the discussion, I will put here some informations about ISLR2. please indicate what is/are the known functions/roles of ISLR2 in general and specifically in the gut. Did mutations/alterations of its expression/localization are associated to pathologies and diseases ?

6- Results section : the way authors showed their data is not appropriate in my opinion. Fig 1 is about overexpression when the result section start with Fig 3 : decrease in ISLR2 expression (by the way Fig 3 is not about effect of ZEN on ISLR2 expression, please check). I will suggest to first show the dose-dependent effect of ZEN on IPEC-J2 viability (I did not see this data that is very important to see if ISLR2 expression effect is specific (hapening in the absence of toxicity) or appears only at toxic doses (suggesting general effect, not specific of ISLR2). So, please first provide such critical data. Then, I will show the dose-effect of ZEN on ISLR2 expression. Then I will show the preventing effect of overexpression of ISLR2 on ZEN toxicity. Please correct.

7- Figure 1 : title is not related to what is shown in Fig 1. What is the dose of ZEN used in Fig1A ? This has to be indicated and comment if it is a toxic dose or not based on missing dose-dep effect of ZEN on viability (needed to show). Ideally, dose-dep effect of ZEN on ISRL2 expression should be performed, not just one dose.

8- Fig 2 : what the dose of ZEN used ? Is it toxic at this dose ? Fig 2B : ROS level of untreated cells is missing and it is needed to see if ZEN alone increase ROS or not. If ZEN does not increase ROS, then ISLR2 overexpression could not inhibit a decrease that does not exist...

9-  Fig 3 : same : i) please indicate ZEN dose used, ii) please add data of untreated cells.

10- section 3.4 : I do not see the link between this section and the rest of the study. Although interesting, this section must be removed (may be used in an other study/publication) or connected to ZEN effect (I do not see how to connect this section to ZEN).

Author Response

In this study, the authors investigated the effect of Zearalenone (ZEN) on porcine intestinal cells looking at is the regulators involved in ZEN cellular toxicity in this model. Authors demonstrated a significant decrease in expression of the ISLR2 gene suggesting a role of this gene in ZEN toxicity. Role of ISLR2 was further demonstrated by the authors as when they overexpressed the gene, so protection against ZEN was observed. Finally, looking at expression regulation of ISLR2, the authors found a G/A mutation located in the promoter region decreasing its transcription. Authors predicted that the G/A mutation is affecting the binding of NF-κB. Overall, this study identified ISLR2 gene as a potential novel target of ZEN-induced toxicity and how it can be regulated by NF-kB.

I found the study interesting. However, I have few issues with this manuscript.

Few comments here:

1- in the abstract, the authors used ZEN and ZEA term. Please homogenize.

Response: ZEN was used in our revised manuscript.

2- in the abstract (but please check the full manuscript) please correct « decrease of » to « decrease in ». same for « increase of » to « increase in »

Response: We have checked the full manuscript and corrected “decrease of and increase of” to “decrease in and increase in”.

3- In the abstract: please indicate in the abstract or title the meaning of ISLR2 (when first used: title or first use in abstract)

Response: We have indicated the full name (immunoglobulin superfamily containing leucine rich repeat 2) of ISLR2.

4- in the abstract: please indicate that the porcine intestinal cells used are IPEC-J2.

Response: The abbreviation IPEC-J2 has been indicated in our revised manuscript.

5- introduction: Although present in the discussion, I will put here some informations about ISLR2. please indicate what is/are the known functions/roles of ISLR2 in general and specifically in the gut. Did mutations/alterations of its expression/localization are associated to pathologies and diseases?

 Response: Thanks for your suggestions. ISLR2 interacts with other proteins to regulate the development of gastrointestinal tract, and alterations of ISLR2 expression levels were associated with pseudoexfoliation syndrome. These findings about ISLR2 functions have been provided in our revised manuscript (lines 92-94 on page 4).

 6- Results section: the way authors showed their data is not appropriate in my opinion. Fig 1 is about overexpression when the result section start with Fig 3: decrease in ISLR2 expression (by the way Fig 3 is not about effect of ZEN on ISLR2 expression, please check). I will suggest to first show the dose-dependent effect of ZEN on IPEC-J2 viability (I did not see this data that is very important to see if ISLR2 expression effect is specific (hapening in the absence of toxicity) or appears only at toxic doses (suggesting general effect, not specific of ISLR2). So, please first provide such critical data. Then, I will show the dose-effect of ZEN on ISLR2 expression. Then I will show the preventing effect of overexpression of ISLR2 on ZEN toxicity. Please correct.

Response: Thanks for your suggestions. We have shown the dose-dependent effect of ZEN on IPEC-J2 viability in Fig. 1A and the dose-effect of ZEN on ISLR2 expression in Fig. 1B.

7- Figure 1: title is not related to what is shown in Fig 1. What is the dose of ZEN used in Fig1A? This has to be indicated and comment if it is a toxic dose or not based on missing dose-dep effect of ZEN on viability (needed to show). Ideally, dose-dep effect of ZEN on ISRL2 expression should be performed, not just one dose.

Response: Thanks for your suggestions. We have investigated the effects of different dose of ZEN on cell viability and ISLR2 expression in Fig. 1.

8- Fig 2: what the dose of ZEN used? Is it toxic at this dose? Fig 2B: ROS level of untreated cells is missing and it is needed to see if ZEN alone increase ROS or not. If ZEN does not increase ROS, then ISLR2 overexpression could not inhibit a decrease that does not exist.......

Response: The dose of ZEN (10 ug/ml) has been indicated in all figures. ROS level of untreated cells has been provided in Fig. 2B, which indicated that ZEN alone can increase ROS level.

9- Fig 3: same: i) please indicate ZEN dose used, ii) please add data of untreated cells.

Response: The dose of ZEN (10 ug/ml) has been indicated. The data of untreated cells has been provided.

10- section 3.4: I do not see the link between this section and the rest of the study. Although interesting, this section must be removed (may be used in another study/publication) or connected to ZEN effect (I do not see how to connect this section to ZEN).

Response: Thanks for your suggestions. In this study, we found significant decreases in the expression of ISLR2 upon ZEN exposure, and ISLR2 overexpression reduced the cytotoxicity of ZEN. According these findings, we want to know which factors may regulate the expression of ISLR2. Therefore, in section 3.4, we identified the genetic variants in the promoter region which are known factors involved in the regulation of gene expressions. The genetic variants can be linked to ZEN toxic effects by regulating the expression level of ISLR2.

Round 2

Reviewer 1 Report

Still, there does not seem to be sufficient evidence to determine whether the G/A mutation affects NF-kB binding. However, the observations in section 3.4 provide some support for the authors' speculation and are worth describing.

Author Response

Still, there does not seem to be sufficient evidence to determine whether the G/A mutation affects NF-kB binding. However, the observations in section 3.4 provide some support for the authors' speculation and are worth describing.

Response: Thanks for your comments. After careful considerations of your and other reviewers’ suggestions on this part, we removed the results of G/A mutation effect analysis in our revised manuscript. We will further fully explore the regulatory role of this mutation on gene expression and link its direct connections with ZEN effects by in vivo and in vitro assays, and report the findings in another publication.

Reviewer 3 Report

Dear Authors,

Thank you for your corrections.

I still have two issues/comments:

1- In Figure 1 showing dose-effect of ZEN, the X-axis has no label, please add it (should be [ZEN] (µg/ml) I suppose)

2- In my first evaluation report, I indicated:

"10- section 3.4: I do not see the link between this section and the rest of the study. Although interesting, this section must be removed (may be used in another study/publication) or connected to ZEN effect (I do not see how to connect this section to ZEN).

Your Response: Thanks for your suggestions. In this study, we found significant decreases in the expression of ISLR2 upon ZEN exposure, and ISLR2 overexpression reduced the cytotoxicity of ZEN. According these findings, we want to know which factors may regulate the expression of ISLR2. Therefore, in section 3.4, we identified the genetic variants in the promoter region which are known factors involved in the regulation of gene expressions. The genetic variants can be linked to ZEN toxic effects by regulating the expression level of ISLR2."

I still don't see any connexion between the rest of the study/manuscript and this part. Except if you show that ZEN activates NFkB in your system and that ZEN effect on ISLR2 expression is prevented by inhibitor of NFkB, I will recommend to simply remove this part and keep it for another publication.

Regards

Author Response

I still have two issues/comments:

1- In Figure 1 showing dose-effect of ZEN, the X-axis has no label, please add it (should be [ZEN] (µg/ml) I suppose)

Response: Thanks for your comments. We have added the label of X-axis in Figure 1.

2- In my first evaluation report, I indicated:

"10- section 3.4: I do not see the link between this section and the rest of the study. Although interesting, this section must be removed (may be used in another study/publication) or connected to ZEN effect (I do not see how to connect this section to ZEN).

Your Response: Thanks for your suggestions. In this study, we found significant decreases in the expression of ISLR2 upon ZEN exposure, and ISLR2 overexpression reduced the cytotoxicity of ZEN. According these findings, we want to know which factors may regulate the expression of ISLR2. Therefore, in section 3.4, we identified the genetic variants in the promoter region which are known factors involved in the regulation of gene expressions. The genetic variants can be linked to ZEN toxic effects by regulating the expression level of ISLR2."

I still don't see any connection between the rest of the study/manuscript and this part. Except if you show that ZEN activates NFkB in your system and that ZEN effect on ISLR2 expression is prevented by inhibitor of NFkB, I will recommend to simply remove this part and keep it for another publication.

Response: Thanks for your comments. According to your suggestions, we have removed this part in our revised manuscript, and we will investigate the regulatory role of the promoter genetic variants on ISLR2 gene expression and reveal the underlying molecular mechanisms in our future study.